# Temporary and Permanent Auditory Effects Associated with Occupational Coexposure to Low Levels of Noise and Solvents

**DOI:** 10.3390/ijerph19169894

**Published:** 2022-08-11

**Authors:** Vanessa Bohn, Thais C. Morata, Simone Roggia, Fernanda Zucki, Benoît Pouyatos, Thomas Venet, Edward Krieg, Maria Renata José, Adriana B. M. de Lacerda

**Affiliations:** 1Post Graduate Program of Communication Disorders, University Tuiuti of Paraná, Curitiba 82010-330, Brazil; 2National Institute for Occupational Safety and Health (NIOSH), Cincinnati, OH 45226, USA; 3Department of Audiology and Speech Therapy, Federal University of Santa Catarina, Florianopolis 88030-300, Brazil; 4Institut National de Recherche et Sécurité (INRS), 54519 Vandoeuvre-lès-Nancy, France; 5Audiology Department, Speech Language and Audiology School, Medicine Faculty, University of Montréal, Montreal, QC H3C 3J7, Canada

**Keywords:** hearing loss, hearing fatigue, ototoxicity, chemical, prevention

## Abstract

This study aimed to assess temporary and permanent auditory effects associated with occupational coexposure to low levels of noise and solvents. Cross-sectional study with 25 printing industry workers simultaneously exposed to low noise (<80 dBA TWA) and low levels of solvents. The control group consisted of 29 industry workers without the selected exposures. Participants answered a questionnaire and underwent auditory tests. Auditory fatigue was measured by comparing the acoustic reflex threshold before and after the workday. Workers coexposed to solvents and noise showed significantly worse results in auditory tests in comparison with the participants in the control group. Auditory brainstem response results showed differences in III–V interpeak intervals (*p* = 0.046 in right ear; *p* = 0.039 in left ear). Mean dichotic digits scores (exposed = 89.5 ± 13.33; controls = 96.40 ± 4.46) were only different in the left ear (*p* = 0.054). The comparison of pre and postacoustic reflex testing indicated mean differences (*p* = 0.032) between the exposed (4.58 ± 6.8) and controls (0 ± 4.62) groups. This study provides evidence of a possible temporary effect (hearing fatigue) at the level of the acoustic reflex of the stapedius muscle. The permanent effects were identified mainly at the level of the high brainstem and in the auditory ability of binaural integration.

## 1. Introduction

Workers in the printing industry are commonly exposed to solvents and noise [1]. Epidemiologic studies have indicated an association between such exposures and hearing dysfunction [2,3,4]. Similar findings for other occupational activities have revealed damage to the peripheral auditory system [5,6,7], to the acoustic stapedial reflex [2,7,8], to the binaural integration ability [9,10,11,12], and to the brainstem auditory pathways [7,13,14]. The preponderance of the evidence suggests solvent effects disrupt peripheral and central auditory functions [10,11,15].

Although pure tone audiometry is an important test for the annual monitoring of workers, it may not be sufficient to detect early signs of hearing loss in workers exposed to chemicals [7,8,10,12,15,16,17,18,19]. Several authors have recommended acoustic reflex measurement as an important tool to differentiate the effects of exposure to noise and solvents and auditory evoked potentials to detect solvent neurotoxicity in the auditory system before changes in pure tone audiometry are detected [2,7,20]. Early detection could help prevent hearing loss caused by the effects of this coexposure [15].

Before occupational hearing loss occurs, early signs and symptoms can be identified, such as hearing fatigue, difficulty understanding in environments with competitive noise, and tinnitus [21]. Hearing fatigue is recognized as an indication of overexposure. It is defined as the temporary decrease in hearing ability; after cessation of exposure and hearing rest, the hearing gradually recovers. It is considered an early symptom in the development of occupational hearing loss, which can be a consequence of long-term hearing fatigue [21,22,23]. The literature suggests that repeated occurrences of hearing fatigue are associated with permanent auditory deficits [21,22,23].

Authoritative documents which reviewed the evidence available to facilitate the proposal of guidance to the occupational health community have concluded that the effects of workplace chemicals on the auditory system can be diverse [15,18,19]. They have indicated that most of the studies conducted to date make use of different audiological tests in the evaluation of their ototoxicity. Cases of hearing loss induced by solvents measured by pure-tone audiometry range from moderate to severe, as do cases of noise-induced hearing loss (NIHL), and they often share the high-frequency audiometric notch configuration. A few reports have indicated that broader frequency bands are affected following solvent exposures when compared to those affected by noise [18].

The preponderance of the evidence indicates that some chemicals affect not only the sensory organ of the auditory system (the cochlea), but also lead to adverse effects on the central auditory structures [10]. Clinical studies have suggested that exposure to certain industrial chemicals may have retrocochlear effects [9,10,11,12,13], and that they can modify the effects of noise [2,16,20]. Chemicals such as solvents, pesticides, and metals have both ototoxic and neurotoxic properties [9]. Audiological signs of neurotoxicity may or may not include poor auditory thresholds, in addition to difficulties in discriminating sounds (such as speech) mainly in adverse listening conditions [9].

Because of the ubiquity of their use in the workplace, solvents are one of the most studied class of chemicals for their toxicity. Millions of workers around the world are exposed to organic solvents such as toluene and xylene in several industrial sectors [6,12,17]. Solvents are neurotoxic substances that are detrimental to the functioning of the nervous system, including the central auditory nervous system. Since they are lipolytic, they have a high affinity for lipid-rich tissues such as brain tissue [18,19,20]. Thus, exposure to solvents is associated with both cochlear auditory dysfunctions and central auditory dysfunctions. Hitherto, these effects were reported to be permanent, given their study designs and the clinical tests used [20]. Studies on whether solvent exposures are associated with temporary effects are needed [6]. 

The Echoscan device [24] was designed to objectively assess auditory fatigue (a temporary effect) by measuring the change in the acoustic reflex threshold following noise exposure. The Echoscan uses a new technology, and its operation is detailed in earlier publications [25,26]. It uses the contralateral suppression of otoacoustic emissions by the triggering of the middle ear acoustic reflex [25,26]. This equipment can be used to identify work environments which can cause auditory fatigue. It is the first time this device has been field-tested with solvent-exposed workers. Its use could facilitate the tracking of early signs of cochlear and middle ear impairment and assess the impact of hazardous exposures on the acoustic reflex. It could facilitate the implementation of early preventive actions [23,26]. In a pilot study with call-center workers, the Echoscan was used to objectively verify whether this population presented hearing fatigue at the end of their work shift (when they were exposed to 75 dBA). The participants had tonal threshold audiometry and the Echoscan testing before and after the same workday. At the conclusion of the study, peripheral and central hearing fatigue was not identified after daily exposure to low intensity noise [26]. 

There is no consensus on clinical audiology protocols to identify early cases of solvent-induced hearing dysfunction and provide hearing care for those with normal audiometry who experience listening difficulties following combined noise and solvent exposure. Studies have suggested the use of a peripheral and central test battery to identify early cases of occupational hearing impairment [20,27,28].

Th present study sought to answer the three questions: Are low levels of noise and solvent coexposure harmful to hearing? If so, are these temporary or permanent effects? Which audiological tests are the most sensitive to detect the early signs of these effects? The aim of this study was to assess the temporary or permanent auditory effects associated with occupational coexposures to low levels of noise and solvents in printing facilities using an auditory test battery to identify early signs of hearing dysfunction.

## 2. Materials and Methods

The present study used a cross-sectional design. It was approved by the Research Ethics Committee of the Universidade Tuiuti do Paraná under number 3.401.602 and CAEE: 10758719.5.0000.8040. All procedures complied with the ethical principles of Resolution 466/2012 of the National Health Council of Brazil. Data produced in this study were analyzed in accordance with a prespecified analysis plan detailed in the documentation for the University’s review board. 

### 2.1. Study Sites

Workers from two printing companies located in southern Brazil were selected to participate in the study. Both companies manufacture and assemble high-quality calendars, catalogs, packaging, folders, stationery products, files, and magazines (Figure 1). Both companies have an Environmental Risk Prevention Program and an Occupational Health Surveillance Program. These programs are recommended at a national level by the Ministry of Labor through its regulatory standards. A walkthrough survey of noise and solvent exposure was conducted for comparison with the industrial hygiene reports of the studied companies.

Data collection took place in two phases. Phase I was carried out on the companies’ facilities to measure auditory fatigue. Phase II was carried out at the audiology clinic of IELUSC University to diagnose permanent effects on hearing. The studied printing industries had similar physical facilities, production processes, and exposure scenarios, which were demonstrated by the document analysis performed in Phase I. This enabled us to combine the data for the analysis. The protocol and aims of the study were explained to representatives of the two companies. Agreement letters were signed, and participant selection followed.

### 2.2. Recruitment and Inclusion Criteria

The study groups were selected using a convenience sampling [29], with participants being recruited at the companies’ facilities. The following inclusion criteria were adopted: working for more than six months in the company; presenting average hearing thresholds (500, 1000, 2000, and 4000 Hz) ≤ 25 dB HL (WHO, 2019); a tympanometry curve without middle ear alterations [30]; no history of use of ototoxic drugs such as antibiotics, diuretics, and aspirin; agreeing to participate to the study and signing a free and informed consent form; and being exposed to a concentrations of solvent below the exposure limits established by the Brazilian legislation [31]. These are 78 ppm (340 mg/m^3^) for ethylbenzene, 78 ppm (340 mg/m^3^) for xylenes, 78 ppm (290 mg/m^3^) for toluene, and 50 ppm (176.24 mg/m^3^) for n-hexane [32] for a 48 h work week. The Brazilian exposure limit for noise is 85 dB(A) time-weighted average (TWA), using a 5 dB exchange rate [31]. The only exclusion criterion used was the presence of impediments in the external acoustic canal (excess earwax or presence of foreign body) observed during meatoscopy, which would have prevented the completion of some of the hearing tests. 

### 2.3. Participants

Fifty-four workers participated in the study; twenty-five of them were exposed to low levels of noise and solvents (exposed group, EG) and twenty-nine were not exposed (control group, CG). Out of the 25 exposed workers, 10 participated in phase I, 8 in phase II, and 7 participated in both phases (see Figure 2). Participants from the control group had different occupations in administrative, commercial, educational, business, and healthcare areas. Out of the 29 participants in the control group, 13 participated in phase I, 12 in phase II, and 4 participated in both phases. Failure to meet the eligibility criteria and difficulty to access the testing location at available times were the reasons some participants were unable to complete one or both study phases (Figure 2). Due to the limited availability of equipment and difficulties associated with testing at a remote location, phase II could only be carried out with participants from company B. 

Figure 2 represents the flowchart of participants at each study phase. Sixty-five volunteers were excluded from the study for presenting middle ear alterations, average hearing thresholds (500, 1000, 2000, and 4000 Hz) > 25 dB HL, and absence of otoacoustic emissions and/or acoustic reflexes. Of the 65 individuals who volunteered to be in the study but did not meet the eligibility criteria, 18 were unexposed controls who were excluded for presenting middle ear alterations, average hearing thresholds (500, 1000, 2000, and 4000 Hz) > 25 dB HL, and/or previous exposure to noise and/or solvents.

### 2.4. Audiological Assessment

The companies’ audiometric records were examined to identify those who presented average hearing thresholds (500, 1000, 2000, and 4000 Hz) below 25 dB HL. Those who met that criterion were asked to answer a questionnaire that gathered information on previous occupational and nonoccupational exposure, hearing symptoms, family history of hearing loss, use of individual hearing protectors, general health, and drug use, among other health information. As a follow-up, workers underwent an inspection of the external auditory canal to check for any impediments to the performance of an audiological assessment (Heine otoscope, model mini 3000), acoustic immittance testing, and the assessment of auditory fatigue (Phase 1). Data collection took place in a quiet room on the companies’ facilities, which included a portable audiometric booth. Hearing examinations were performed at least 14 h after the last exposure to noise to minimize the detection of temporary threshold shifts. On the same day, after an average of six hours after the beginning of their work shift Echoscan measurements were repeated to diagnose the auditory fatigue. With one exception, all participants from the exposed group reported using hearing protectors (earplugs with an approximate noise attenuation rating of 16 dB) throughout the work shift.

Phase 2 consisted of completing the test battery at the Audiology Clinic School, with the following procedures: acoustic immittance, auditory brainstem response (ABR), and the dichotic digits test (DD), as described in Table 1. All equipment were calibrated before data collection and biological calibration was performed each day before subjects were tested.

The volunteers whose responses were not in the normal range on some tests were referred for hearing monitoring or consultation with a specialist. 

### 2.5. Assessment of Solvent Concentration

A documentary analysis of the chemical risk assessment conducted by the companies’ environmental risk prevention program was carried out. The reports described measurements conducted by an external contractor through individual sampling of representative subgroups of exposed workers in each of the participating companies. For the study participants who did not participate in the sampling, the mean value of the solvent concentrations evaluated for the same job and same work task was used to evaluate exposures. This analysis guided the selection of study participants from the printing, finishing, shipping, continuous improvement, and processing sectors. The results of the report indicated much lower time-weighted average exposure levels to solvents than the Brazilian exposure limits for each of the detected solvents. Passive samplers were also used to collect full-shift air samples in a subsample of each of the groups of workers who had the same overall exposure profile for the agent(s) under study. The sampling device was positioned as close as possible to the breathing zone of the worker (carried by the worker). The adsorption tube samples were sealed and mailed to the U.S. for gas chromatography. These measurements were carried out on the day of the hearing tests. 

### 2.6. Noise Assessment

The information on noise dose, according to each participant’s sector and function, was obtained from the companies’ 2018 risk assessment reports. Noise levels ranged from 57 dB(A) to 83 dB(A). These levels were confirmed by a walk-through survey by the investigator team using the NIOSH Sound Level Meter app. All workers were exposed to noise levels below the limits established by NR15 of ≤85 dB(A) [31]. Despite the time-weighted average not surpassing the 85 dB(A) limit, which triggers mandatory hearing protection, both companies required their use, considering that in some instances noise *levels* were actually higher than 85 dB(A).

### 2.7. Statistical Methods

The results of the hearing tests as continuous variables were analyzed with linear models assuming normality. An arcsine transformation was used for the measurements from the dichotic digits test. Dichotomous outcomes were analyzed with a logistic regression using exact tests and confidence intervals. Linear and logistic models were performed to estimate and test the effects on the exposure group. In addition to the exposure group, the models included variables for study, age, sex, previous noise exposure, and previous chemical exposure. A second set of models was performed with two exposure variables, noise level and current chemical exposure, replacing the exposure group. The same covariates were included. Adjustments were made for age and gender within the models, not just matching between groups. The adjustments were made even if there was no statistical significance. All calculations were performed using SAS^®^ Version 9.4 (SAS Institute, Inc., Cary, NC, USA). PROC MIXED was used for the linear models. PROC GENMOD was used for the logistic regression models.

## 3. Results

Table 2 shows the characterization of the sample, followed by the results of each of the auditory tests. The study population had a high proportion of female workers; gender was not found to be associated with the outcomes of any of the tests in the battery.

Information on previous occupational and nonoccupational exposure, hearing symptoms, family history of hearing loss, use of individual hearing protectors, general health, and drug use was not significantly different among groups. Despite not being different between groups, we kept these variables in the model when testing for correlations. Significant (or quasi-significant) correlations were only observed between PTA results of the exposed group and the reported use of hearing protectors. 

The solvent concentration assessment results obtained from the measurements taken on the day of the hearing tests were in accordance with the results described in the risk assessment reports obtained from the participating companies. The levels of ethylbenzene ranged from 3.6 ng to 61 ng, n-hexane from 4.0 ng to 850 ng, xylenes from 3.5 ng to 53 ng, and toluene from 4.4 ng to 77 ng. The results were well below both the Brazilian exposure limits [31] and the levels recommended by the American Conference of Governmental Industrial Hygienists (ACGIH) [38]. For an illustration of solvent and noise exposure scenarios in printing companies, see Alabdulhadi et al. [39,40]

### 3.1. Signs and Symptoms

No significant differences were observed in the rates of hearing complaints between the groups. The overall rates among the 54 participants were as follows: 60% (*n* = 15) had at least one auditory and/or extra-auditory complaint, out of which 36% (*n* = 9) had itching, 24% (*n* = 6) reported discomfort with intense sound, 16% (*n* = 4) complained of otalgia, 12% (*n* = 3) had a feeling of ear fullness, 12% (*n* = 3) had difficulty in understanding speech in a noisy environment, 12% (*n* = 3) had tinnitus, and 8% (*n* = 2) had vertigo. It is important to note that all participants had average thresholds < 25 dB HL.

### 3.2. Pure Tone Audiometry

Figure 3 shows the comparison of the average hearing thresholds (500–8000 Hz) for the right and left ears, by exposure group. Poorer results were observed for the exposed group for the frequency of 6000 Hz in the right ear (control group’s mean = 12.75 dB ± 7.62 and exposed group’s mean = 16 dB ± 7.5; *p* = 0.022) and 4000 Hz for the left ear (control group’s mean = 6.72 dB ± 8.15 and exposed group’s mean = 10 dB ± 6.77; *p* = 0.010).

The results of the linear and logistic models created to estimate and test for the effect of the two exposure variables, noise level and current chemical exposure (replacing exposure group), indicated a significant difference was observed in the results at the frequency of 4000 Hz for the left ear and the frequency of 6000 Hz for the right ear. When we tested for correlations among the variables included in the model, hearing protector use in the exposed group was inversely related to some pure tone audiometry thresholds, PTA right 3000 Hz (r = −0.39242, *p* = 0.0523), PTA left 2000 Hz (r = −0.36690, *p* = 0.0712), PTA left 3000 Hz (r = −0.51450, *p* = 0.0085), PTA left 4000 (r = −0.60302, *p* = 0.0014), and PTA left 6000 Hz (r = −0.37330, *p* = 0.0661). 

### 3.3. Immittance Audiometry—Thresholds of Acoustic Reflexes 

A between-group comparison of the intensities needed to trigger the acoustic reflex (dB HL) obtained through immittance audiometry only detected a significantly lower level at the contralateral stimulation at the right ear at 2000 Hz for the exposed group (*p* = 0.059), when the exposure group was in the model. The results of the linear and logistic models developed with two exposure variables, noise level and current chemical exposure levels, showed a statistically significant difference for noise at the frequencies of 500 Hz (*p* = 0.024) and 1000 Hz (*p* = 0.043) for the ipsilateral stimulation of the right ear and at 500 Hz for the ipsilateral stimulation of the left ear (*p* = 0.049). 

### 3.4. Echoscan Assessment of Auditory Fatigue

The assessment of auditory fatigue using the Echoscan equipment was calculated through the difference of acoustic reflex measurement before and after their work shift (see Figure 4). The mean post-and-pre work shift response variation for the controls was 0.0 dB ± 4.62 while the exposed group had a mean response variation of 4.6 dB ± 6.8 (classified as possible hearing fatigue). Mean comparisons revealed a significant difference between the groups (*p* = 0.031). The risk estimate adjusted for noise and chemical exposure as variables revealed a significant difference (*p* = 0.010) between the exposed versus nonexposed group.

### 3.5. Dichotic Digits Test

Figure 5 shows the participants’ mean dichotic digits responses by ear. The mean score for the exposed participants was 91% ± 20 for the right ear and 90% ± 13 for the left ear. The mean scores for the controls were 98% ± 3 for the right ear and 96% ± 5 for the left. The comparison of the mean results by group and by ear revealed a nearly significant difference observed for the left ear (*p* = 0.054). In the estimation of risk in the adjusted models with noise level and chemical exposure as exposure variables, a trend was observed for the difference between groups (*p* = 0.064).

### 3.6. Auditory Brainstem Response—ABR

A significant difference was only observed for the ABR interpeak interval of the III–V waves for the right ear (mean latency 1.93 ± 0.1 ms among controls and 2.00 ± 0.12 ms among exposed; *p* = 0.046) and in the left ear (mean latency 1.94 ± 0.14 ms among controls and 2.01 ± 0.13 ms among exposed; *p* = 0.039) (Table 3).

## 4. Discussion

This study aimed to assess temporary and permanent auditory effects associated with occupational coexposure to low levels of solvents and noise, using an auditory test battery, and the results showed it can be harmful to hearing, causing temporary and permanent effects. The temporary effect was identified at the level of the acoustic reflex of the stapedius muscle, which may be an early sign of hearing impairment. The permanent effects were identified mainly at the level of the high brainstem and in the auditory ability of binaural integration. The results of the Echoscan, auditory brainstem response (ABR), and dichotic digit tests (DD) indicated that these were the most sensitive tests in the early identification of auditory effects, both temporary and permanent.

### 4.1. Exposure Duration, Age, Gender, and Symptoms

The studied population was young (mean age below 37 years, age-matched controls), had short tenures at the studied companies (5–7 years), and the volunteers were predominantly female. Neither of these variables was associated with the studied outcomes. An association with age was not expected, as our control group was matched by age to the exposed group and the logistic regression model adjusted for that. Regarding gender, the findings of this study do not fully agree with prevailing evidence. Female workers have been considered less vulnerable to the effects of noise [41], or it has been argued that the difference would be explained by different noise exposure trajectories throughout their lives [42]. In the present study, gender was not significantly associated with any of the test results, but it is possible that our sample size did not allow the detection of such differences. Similarly, there were only a few subjects in which the overall exposure duration, in the previous or current job, was associated with the hearing test results. Still, it is concerning that test results differed between exposed and nonexposed, given their low exposures and short tenures. Current evidence does not offer answers on whether the solvent effects observed in humans were caused by long-term exposure to low-level background exposure to solvents (expressed as time-weighted averages) or if they were triggered by a few peaks of high solvent concentration. It is conceivable that peaks of solvent exposure are sufficient to trigger some damage and explain the observed effects. This question could only be answered in an investigation with a larger sample size and detailed noise and chemical exposure assessments.

Although auditory symptoms such as itching (likely to be explained by the earplug use), discomfort caused by loud sounds, otalgia, sensation of ear fullness, difficulty in understanding speech in a noisy environment, tinnitus, and vertigo were reported, no significant differences were observed between groups. Many of these specific complaints related to hearing dysfunction can be considered as preclinical, occurring prior to test results that would make the dysfunction explicit [15,20].

### 4.2. Pure Tone Audiometry

Although the hearing threshold means were within normal levels and the exposed group was exposed to low levels of noise and solvents, significant differences were observed at 6000 Hz in the right ear and 4000 Hz in left ear in the group comparisons (Figure 3), with poorer results from the exposed group. These findings corroborate the findings of earlier studies [4,7,12,13,43,44,45,46]. The subclinical findings observed may be associated with other confounding variables (other nonoccupational exposures) [20]. The hearing thresholds of the exposed group were found to be correlated with hearing protection use. It is important to note that in both studied companies the use of hearing protection is optional, and a matter of individual preference, given that their noise exposures, measured as time-weighted averages, do not trigger the requirement of their use. Observations conducted as part of the walk-through noise and solvent survey suggested that the use of hearing protection was not consistent throughout the work shift, but the data suggested it offered some level of protection to the studied group. 

### 4.3. Immitance Audiometry—Thresholds of Acoustic Reflexes

A significant difference between groups (with worse results from the exposed group) was only observed for the acoustic reflex at 2000 Hz in the right ear, by contralateral stimulation. Roggia et al. [7] also observed a significant difference at the frequency of 2000 Hz in the contralateral acoustic reflex of the right ear, when comparing a control group and a group exposed to fuels, although both groups had hearing thresholds in the normal range. Zucki et al. [8] and Tochetto et al. [44] conducted studies with workers exposed to gasoline (with normal hearing thresholds) and reported altered results in the acoustic reflex both in the ipsilateral and contralateral stimulation, although the differences were not statistically significant. An experimental study by Valero et al. [47] showed that the triggering of the middle ear acoustic reflex in mice was altered in cochlear regions that suggested a synaptopathy. Valero et al. [48] also observed that mice, which have a reduced number of synapses after exposure to noise but do not present inner or outer hair cell loss in the cochlear region, had increased middle ear acoustic reflex thresholds and reduced amplitudes suggesting that the acoustic reflex may be valuable in the early detection of cochlear neuropathy.

### 4.4. Echoscan Assessment of Auditory Fatigue

In the analysis of auditory fatigue, the group coexposed to noise and solvents presented a difference in response between post- and pre-exposure acoustic reflexes thresholds (4.58 dB ± 6.8), indicating possible auditory fatigue (Figure 4). The Echoscan test was designed to measure temporary changes in the threshold of the acoustic reflex to identify work situations that cause auditory fatigue [25]. Campo et al. [49] reported that exposure to aromatic solvents can alter the function of the efferent pathways, since they inhibit the antagonist effects of cholinergic receptors in the brainstem and prevent the triggering of the acoustic reflex. Venet et al. [50] identified that toluene can weaken the system that triggers the acoustic reflex in rats. The lower concentration of toluene increased the reflex amplitude (strengthened the reflex), while the higher dose decreased it (weakened the reflex). In real life, exposures would be closer to the “lower” concentrations used in the animal experiments. Campo et al. [51] reported similar results for exposure to styrene and impact noise, suggesting that the rapid pharmacological effect caused by the exposure to solvents has the potential to be an early marker of solvent ototoxicity, especially when combined with exposures to complex noise. Rats coexposed to styrene and continuous noise had less hearing loss than rats exposed to continuous noise alone. The low dose of styrene strengthened the reflex and thus reduced the damage in coexposed animals. It was not the case with impulse noise, for which the “classic” synergistic effect was observed because the reflex is not protective in this case (as the noise energy from impulse noise enters the cochlea before the reflex is triggered).

Earlier animal experiments have shown that when low concentrations of solvents were present, noise exposure had to reach levels above 80–85 dB for the detection of an increase in reflex thresholds [25,51]. Exposures to solvents alone tended to decrease the threshold of the reflex during the workday (negative shift), which was observed in studies conducted with experimental animals and workers [51]. The fatigue detected in the present experiment (increase in reflex thresholds) would then be more likely to be explained by noise exposures. The information we had on their solvent and noise exposure was limited to time-weighted averages, so we cannot rule out potential peaks of either of the exposures. Without individual exposure assessment conducted on the day of the hearing tests, one cannot be certain. Still, there is evidence of an elevation of the threshold of the acoustic reflex triggering reflecting auditory fatigue despite low exposures well below the regulatory limits.

In the present study, workers coexposed to noise and solvents within regulatory exposure limits displayed auditory fatigue at the end of their workday, which did not occur in the control group. Venet et al. [50] used the Echoscan to examine the hearing of rats using distortion-product otoacoustic emissions before, during, and after the activation of the acoustic stapedius reflex with a contralateral and/or ipsilateral suppression. The results showed that toluene can change the efficiency of the acoustic stapedius reflex, depending on the toluene concentration and on the ear that was receiving the suppressing noise. Venet et al. [50] also showed that the structures involved in the activation of the acoustic reflex were much more complex than previously described by other authors, that is, the authors observed the involvement of interneurons located between the cochlear nuclei and the superior olivary complex. The hypothesis is that these can interpret different afferent responses coming from the two ears, generating a single integrated response to trigger the middle ear acoustic reflex. Thus, it can be suggested that the responses of the auditory system structures assessed by the Echoscan were altered due to changes in the structures involved in the acoustic reflex arc of the stapedius muscle, as can be seen in other tests, such as the ABR. Although the Echoscan results are not specific to solvent exposures, it can help to identify temporary effects on the hearing of workers and identify those in need of preventive interventions. 

### 4.5. Dichotic Digits Test

In the dichotic digits test, worse results were observed in the exposure group (Figure 5), with a nearly significant difference being observed for the left ear compared to the control group, suggesting alterations in central auditory functions. A value close to the significance level (*p* = 0.064) was observed in the analysis of the exposure effect. This finding is in agreement with those from Fuente et al. [9], Fuente and McPherson, [10], Fuente et al. [11,13], Landry and Fuente [52]. Fuente et al. [11] suggested that the dichotic digits test could be an important tool to detect central auditory disorders associated with exposure to solvents. It is also noteworthy that the averages of the results obtained in both ears of the participants in the exposure group were poorer than the threshold of normality used to classify test results [37].

### 4.6. Auditory Brainstem Response—ABR

A significant difference was only observed in the result of the ABR interpeak interval of waves III-V for both ears (Table 3). When assessing gas station attendants with normal hearing thresholds using the ABR, Quevedo et al. [53] concluded that exposure for a minimum period of three years was associated with detectable changes in the central auditory responses. These alterations included an increase in the absolute latencies of waves, interpeak intervals, and interaural difference. Juárez-Pérez et al. [14] also observed significant differences in the result of the absolute latency of waves III and V and the interpeak intervals between the group exposed to low levels of a solvent mixture compared to the nonexposed group. 

Fuente et al. [13] studied 30 participants exposed to xylene and 30 nonexposed participants and observed a significant difference in the ABR test for the absolute latency of wave V and interpeak intervals IV and III–V for the right ear and in the absolute latency of waves III and V and in the I–V interpeak intervals for the left ear of xylene-exposed individuals compared to nonexposed individuals, with xylene-exposed participants presenting worse results. The authors concluded that xylene was associated with adverse effects on the central auditory system. Thus, they suggested that xylene-exposed workers should be assessed with tests that can detect these effects on the auditory system early.

Roggia et al. [7] and Tunsaringkarn et al. [54] also observed a significant difference between the group exposed to fuel vapors that have benzene, toluene, ethylbenzene, and xylene in their composition and a control group. Roggia et al. [7] observed increased absolute latencies in wave V bilaterally with a significant difference between the group with normal hearing exposed to fuel and the control group. Participants with hearing loss exposed to chemicals had increased III–V and I–V interpeak intervals for both ears compared to the nonexposed group. Fuente et al. [6] observed an association between exposure to aircraft fuel and the ABR absolute latency of wave V in the right ear. 

In our analysis a significant difference was observed between exposure groups for the III–V interpeak interval for both ears. These findings are more likely associated with exposure to chemicals, which have neurotoxic properties, than with noise exposure. The III–V interpeak interval is related to signals generated in the brainstem, demonstrating the conduction between the cochlear nuclei and the superior olive [55]. Thus, the results found in the present study indicate the possibility that workers exposed to low noise and low concentrations of solvents may present a dysfunction in their central auditory system at the brainstem level, corroborating studies by Roggia et al. [7], Juárez-Pérez et al. [14], Fuente et al. [13], and Quevedo et al. [53].

### 4.7. Temporary Effects vs. Permanent Effects

Evaluating temporary effects of occupational exposures to noise on hearing presents significant challenges [56]. Noise exposure time-weighted averages above 80 dBA have been identified as necessary for the triggering of a temporary threshold shift [57]. Ideally, participants are to be tested with 2–6 min following the end of the exposure, which can be extremely difficult to implement [58]. These challenges trigger the need to search for novel approaches to evaluate any temporary effects of the selected exposure scenarios (low exposures to noise and solvents). We selected a test that has been demonstrated in experimental and clinical studies [25,26] to be sensitive to temporary functional changes in hearing, in response to different conditions. Our aims included examining it in the field among groups with different exposure conditions.

The tests used in this study were sensitive to the early detection of temporary as well as permanent changes in the auditory system of printing workers coexposed to low noise and low concentrations of solvents. Some authors have emphasized that performing pure tone audiometry alone is not enough to monitor the negative effects on the auditory system in workers exposed to chemical agents [7,8,18,27]. Studies have recommended that workers exposed to solvents should be monitored with tests that assess the loss of communicative ability, the stapedius reflex, otoacoustic emissions, and ABR [7,13,20,46].

A temporary impairment of hearing capacity after exposure to factors that injure the ears is defined as auditory fatigue. Temporary dysfunction was detected in the exposed group through the Echoscan. Regarding permanent effects, previous studies had suggested that the dichotic digits test [11,12,52], the auditory brainstem response (ABR) [7,13,14,53], and the assessment of the stapedius acoustic reflex [7,8] were sensitive to the otoneurotoxic effect of chemical agents even before the occurrence of hearing loss. 

These tests were developed for clinical use and can identify functional alterations in the auditory pathway of workers coexposed to low noise and low concentrations of solvents. The motivation for including the assessment of hearing fatigue in this study was to examine the occurrence of temporary changes in hearing, from exposure scenarios that are considered safe. We also aimed to evaluate this new tool that could enable early detection and the recognition of workers susceptible to the development of hearing disorders. 

The result of auditory fatigue identified using the Echoscan followed the same trend observed in the response of other auditory tests, that is, poorer results among workers exposed to low noise and low concentrations of solvents in comparison with the unexposed group. 

### 4.8. Implications for Future Studies and Preventive Practices

The present study detected signs of auditory dysfunction that did not seem to be apparent to the individual, as well as to professionals who do not run statistical analyses of group results. Given that exposure scenarios in both companies were considered safe, that workers were young, and had a relative short exposure duration, our findings represent a challenge to prevention programs. Auditory monitoring of these workers seems to be needed, as well as referrals for clinical examinations that assess the peripheral and central auditory pathways.

Investigations which examined the effects of solvents over time indicated that hearing loss was observable two to three years earlier than was usually seen with noise exposure [2,3,16,17,18,19]. Five or more years of noise exposure seem necessary for it to trigger an effect on hearing thresholds following occupational exposures. This issue of latency is certainly dependent on the hazard and the characteristics of the exposure, and needs further investigation [18,19].

Pure-tone audiometry only informs us of the lowest sound that can be perceived in a quiet environment. It only describes some characteristics of the problem but not its potential cause. Information on word recognition difficulties or other hearing problems obtained through a questionnaire can indicate the need for referral to an audiological evaluation when they are inconsistent with pure-tone audiometric results. Moreover, it is important to examine if other hearing testing can complement the pure-tone audiometric assessment conducted with workers. 

The assessment of auditory fatigue by the Echoscan is an objective and quick test, since it does not depend on the worker’s active response, and it can be performed in a quiet room without the need for a soundproof booth. Thus, it has the possibility of being applied in numerous places, such as in companies of different fields, clinics, and hospitals. The Echoscan could facilitate the recognition of the most susceptible groups of workers and the early identification of temporary hearing effects in workers with normal hearing. Ideally, one would be able to monitor exposures to hearing hazards throughout the day, to be able to understand possible causes of the fatigue. This is likely to be hard to implement, and this is not the only potential barrier. The Echoscan evaluation requires workers to be tested before and after their shift. Time away from work is a concern for employers. After-shift testing would most likely require workers staying beyond their workday and could prove difficult to achieve. In addition, many workers already have hearing losses, which would preclude them from being tested with the Echoscan. Even some workers with normal hearing thresholds can have an absence of evoked distortion-product otoacoustic emissions and acoustic reflexes. These need to be present for the completion of the test. 

### 4.9. Study Limitations

An important limitation of this study is the lack of detailed current and previous exposure information of the participants, including data on the individual uptake of solvents. Given its cross-sectional design, we were not able to establish causation, but rather, association between outcome and studied variables. In addition, we used a convenience sample, as a random sample was not possible due to the limited access we had to study populations. Statistical inferences based on a convenience sample only pertain to the sample and not the population it was drawn from, which limits the generalizability of the results to other samples and populations.

While our results indicated that workers in the printing industry have significantly higher rates of peripheral and central auditory dysfunctions than nonexposed persons, it was not possible to identify with precision the contribution that their exposures represented to their hearing status. Many factors can impair the auditory system, several of which were controlled for in this study (age, gender, history of use of ototoxic drugs such as antibiotics, diuretics, and aspirin, previous exposure at other jobs). Studies with larger groups and better exposure information could help shed light on the contribution of different risk factors. In the investigation of temporary auditory effects, it would be ideal to conduct full shift noise and solvents exposure measurements on the day of the testing, identifying possible peaks of exposure, not only their time-weighted averages. 

Another important limitation of this study is that some participants, even with hearing thresholds within normal limits, did not present acoustic reflex responses or distortion-product otoacoustic emissions. Thus, it was not possible to include these participants in the study. Finally, the results of hearing fatigue were based on the standards of the equipment, and it would be informative to functionally verify hearing fatigue using an alternative test.

## 5. Conclusions

Occupational coexposure to low noise and low concentrations of solvents can be harmful to hearing, causing temporary and permanent effects. This study provided evidence for a possible temporary effect, which may be an early sign of hearing dysfunction. The permanent effects were identified mainly at the level of the high brainstem and in the auditory ability of binaural integration. Workers exposed to low concentrations of solvents and noise had significantly poorer performance than nonexposed subjects in pure-tone audiometry, middle ear reflex, dichotic digits, auditory fatigue test, and auditory brainstem response. The results suggest the studied workers had both peripheral and central auditory dysfunctions that could be attributed to their exposures. These results can guide audiologists in the selection of tests to include in a battery for evaluating this population. Finally, the traditional approaches for assessing occupational hearing loss need to be re-examined to allow the early assessment of hearing functions in workers exposed to chemical agents.

## Figures and Tables

**Figure 1 ijerph-19-09894-f001:**
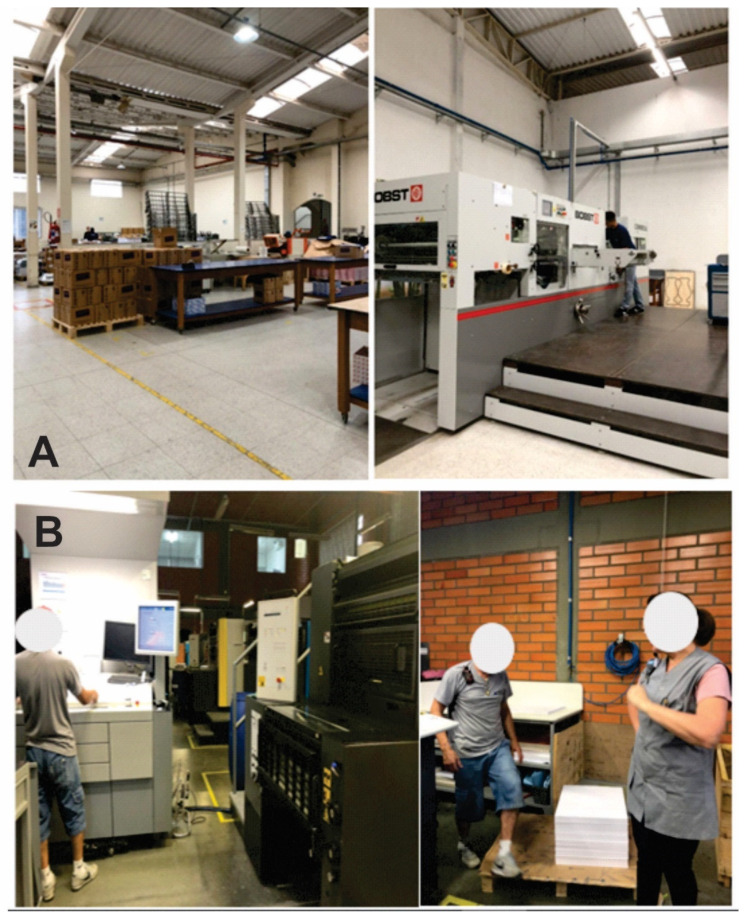
Images of printing equipment and setup locations at each of the companies (**A**,**B**).

**Figure 2 ijerph-19-09894-f002:**
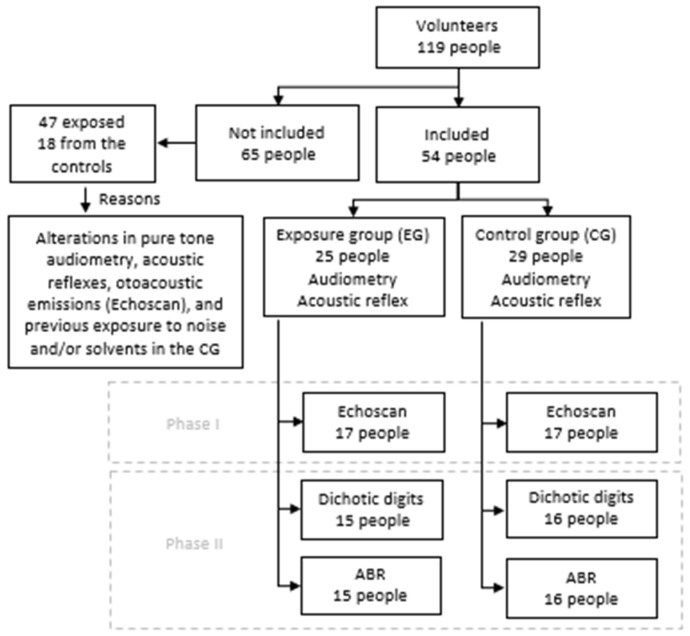
Flowchart of study participants.

**Figure 3 ijerph-19-09894-f003:**
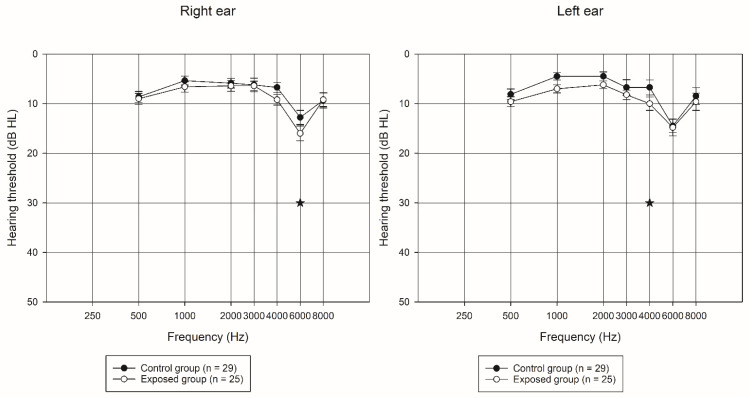
Mean hearing thresholds (+/− one standard error) by ear and exposure group. Star symbol denotes significant differences at the *p* < 0.05 level between study groups.

**Figure 4 ijerph-19-09894-f004:**
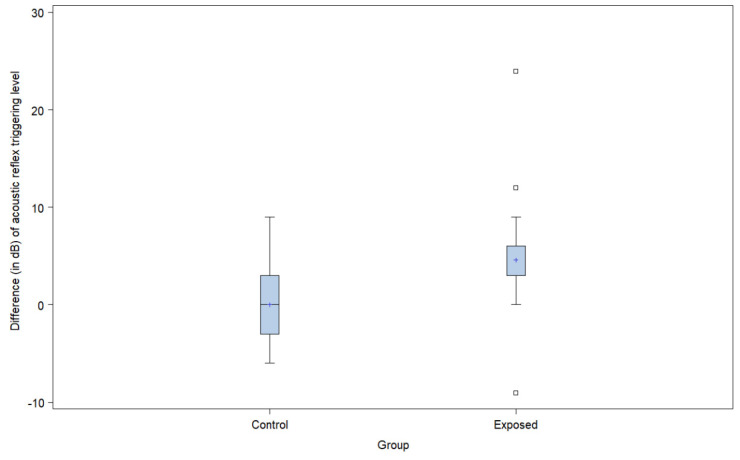
Boxplot of the difference (in dB) of acoustic reflex triggering levels before and after the participant’s work shift, by exposure group.

**Figure 5 ijerph-19-09894-f005:**
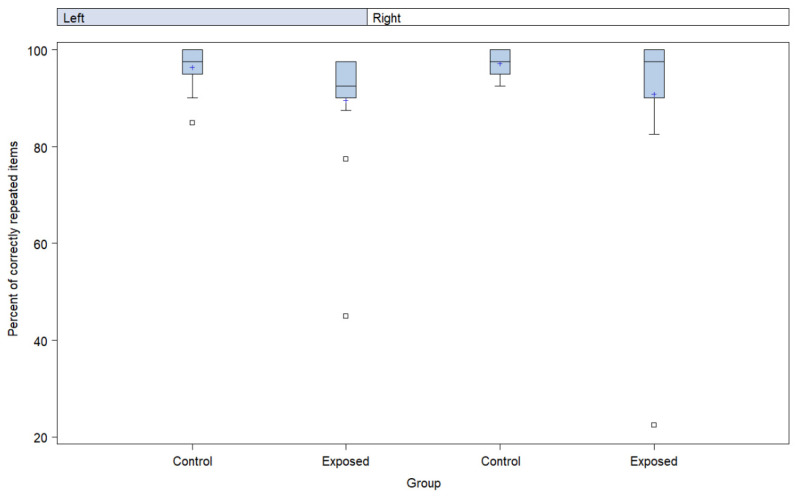
Boxplots of the scores (in percentage) obtained in the dichotic digits test (binaural integration) by exposure group.

**Table 1 ijerph-19-09894-t001:** Audiological test battery used in the investigation.

Procedure	Objective	Equipment and Protocol	Criterion Used for Interpretation
Pure tone audiometry	To determine hearing thresholds by air conduction at the frequencies of 500–8000 Hz and, if necessary, by bone conduction at altered frequencies (500–4000 Hz).	Calibrated audiometer, model Itera II by Madsen^®^ and/or model Ad-229b by Interacoustics, with TDH 39 headphones in a soundproof booth.	≤25 dB HL [33].
Tympanometry	To assess the functioning and integrity of the middle ear.	Middle ear analyzer by Madsen^®^, model Otoflex 100, and/or by Interacoustics, model AT-235 with a 226 Hz probe.	Normal response pattern is compliance values between 0.3 and 1.6 mL and pressure of −100 to 100 daPa [30].
Middle ear reflex	To analyze the integrity of the acoustic reflex pathway.	Middle ear analyzer by Madsen^®^, model Otoflex 100, and/or by Interacoustics, model AT-235, with 226 Hz probe.	For the stapedius reflex thresholds triggered with a sensation level between 70 and 100 dB HL were considered as normal [34,35].
Hearing fatigue assessment (DP-gram, growth curve of DPOAEs, determination of contralateral acoustic reflex)	To assess auditory fatigue and to verify the occurrence of a decrease in the amplitude of distortion-product otoacoustic emissions (DPOAEs) induced by activation of the stapedius reflex	Echoscan^®^ by Echodia. Ipsilateral ear equipped with a probe for DPOAEs measurement with a frequency range f2 between 4000 and 5000 Hz (f1 = 1.2 f2 and an intensity range L2 between 40 and 67 dB (L1 = L2 + 6 dB). Contralateral ear equipped with an earphone producing a band noise (width 800 Hz) centered on 1000 Hz whose intensity varies between 62 and 98 dB HL (3 dB step) to trigger the acoustic reflex. Auditory fatigue is calculated by taking the difference between the thresholds of the auditory reflex before and after exposure.	Pre- and post-test comparison results are classified as follows: result ≥ 9 dB indicates hearing fatigue; result between 3 dB and 6 dB indicates possible hearing fatigue; result ≤ 0 dB does not indicate hearing fatigue [24].
ABR	To assess the integrity of the auditory nerve and the auditory pathway in the brainstem.	Eclipse by EP25 Interacoustics. Electrodes coupled to a preamplifier. Positioning of the electrodes of the 10/20 system: active electrode (Fz); reference electrodes of left ear (A1) and right ear (A2) lobes; ground electrode in Fpz. Individual electrode impedance of less than 5 kW and interelectrode impedance of less than 2 kW. Click stimuli, monaural with a 3 A in-ear earphone, duration of 0.1 millisecond (ms), polarity rarefaction, intensity of 80 dB HL, presentation rate of 27.7 clicks per second, and bandpass filter of 100–3000 Hz. Window of 10 ms, including 1 ms pre-stimulus (electroencephalic EEG activity control). Two records were performed with 2000 stimuli.	The normality criteria adopted for the waves and interpeak intervals were those established for the equipment used: wave I absolute latency (1.14–1.54 ms), wave III absolute latency (3.27–3.71 ms), wave V absolute latency (5.01–5.66 ms), interpeak interval I–III (1.52–2.36 ms), interpeak III–V (1.75–2.07 ms), and interpeak I–V (3.55–4.63). The interaural difference of wave V latency should be a maximum of 0.3 ms [36]. Presence of response is considered when the reproducibility of waves is observed.
Dichotic Digit test—DD	To assess the binaural integration ability	Dichotic digits test (DD) in Portuguese, version published in by Pereira and Schochat 2011. Two-channel audiometer by Madsen, model Itera II, with headphones and a computer coupled to the audiometer. The Brazilian version includes 20 items of two disyllable word pairs presented simultaneously in each ear. Test was performed at 50 dB SL, considering average PTA thresholds at 500, 1000, and 2000 Hz, for each ear.	The reference values adopted were OD ≥ 95% and OE ≥ 95% [37].

**Table 2 ijerph-19-09894-t002:** Characterization of the study sample by exposure group.

VARIABLES	Exposed Group *N* = 25 (*N* = 17, 68% Females)			Control Group *N* = 29 (*N* = 22, 76% Females)		
	Mean	Range	SD	Mean	Range	SD
Age (years)	36.2	17–55	10.8	36.7	17–54	9.4
Tenure (years)	4.8	0.5–20	4.2	7.5	0–21	7.9
Noise TWA dB(A)	75	57–83	7.7	-	-	-
Hearing protector use (%)	80	-	0.4	-	-	-
Previous exposure to noise (%)	52	-	0.5	-	-	-
Previous exposure to chemicals (%)	24	-	0.4	-	-	-

**Table 3 ijerph-19-09894-t003:** Mean (+/− one standard deviation) of auditory brainstem response measurements by ear and exposure group. The asterisk * denotes significant differences at the *p* < 0.05 level between study groups.

	Right Ear			Left Ear		
	Mean ± SD CG (*n* = 16)	Mean ± SD EG (*n* = 15)	*p* Value	Mean ± SD CG (*n* = 16)	Mean ± SD EG (*n* = 15)	*p* Value
Wave I AL (ms)	1.40 ± 0.09	1.41 ± 0.12	0.07	1.39 ± 0.09	1.41 ± 0.08	0.46
Wave III AL (ms)	3.59 ± 0.15	3.60 ± 0.11	0.90	3.59 ± 0.10	3.64 ± 0.13	0.65
Wave V AL (ms)	5.53 ± 0.17	5.60 ± 0.16	0.09	5.53 ± 0.20	5.66 ± 0.20	0.06
I–III (ms) IPI	2.19 ± 0.13	2.18 ± 0.15	0.18	2.19 ± 0.08	2.23 ± 0.13	0.88
III–V (ms) IPI	1.93 ± 0.10	2.00 ± 0.12	0.04 ***	1.94 ± 0.14	2.01 ± 0.13	0.03 ***
I–V (ms) IPI	4.13 ± 0.15	4.19 ± 0.17	0.79	4.13 ± 0.17	4.25 ± 0.20	0.08
Wave amplitude I (µV)	0.27 ± 0.11	0.26 ± 0.08	0.32	0.24 ± 0.09	0.25 ± 0.11	0.68
Wave amplitude III (µV)	0.25 ± 0.12	0.27 ± 0.14	0.52	0.22 ± 0.08	0.24 ± 0.12	0.84
Wave amplitude V (µV)	0.42 ± 0.11	0.43 ± 0.14	0.74	0.38 ± 0.10	0.38 ± 0.12	0.84

## Data Availability

The data presented in this study are available on request from the corresponding author. The data are not publicly available due to privacy and ethical considerations.

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
