# Peer review of "Temporary and Permanent Auditory Effects Associated with Occupational Coexposure to Low Levels of Noise and Solvents"

_ijerph, 2022, doi:10.3390/ijerph19169894_

Round 1

Reviewer 1 Report

This manuscript investigated temporary or permanent auditory effect linked to occupational exposures to low level of noise and solvents in printing facilities. However, the manuscript requires a major revision before it can be accepted for publication. For example, the Introduction did not include a review of other or similar studies that have been reported in the literature, even though the author cited enough of such work in their discussion. Hence, the novelty of this present work is not well justified.

Other revisions:

1.The abstract (results part) should clearly state the findings on how noise and solvents affected the auditory of the workers studied.

The methodology was described in detail, but results were included in section 2.5, which should be removed and included in the results. 

2. At line 91, the caption of Figure 1 should be corrected to .... "A and B", not "A e b".

3. Improve the quality of Figure 2; increase the font size, and restructure it to clearly show the two phases (Phase 1 and Phase 2).

4. Improve the quality of Figure 3; use readable/larger font and improve the quality of the lines too. 

5. Include a section 3.7 on solvent concentration results.

6. On page 12, the paragraph at lines 382-385 should be joined together with the preceding paragraph at lines 386-401.

Author Response

Response letter of manuscript ID: 1815632

 Title: Temporary and Permanent Auditory Effects Associated With Occupational Co-Exposure to Low Levels of Noise and Solvents

Authors: Vanessa Bohn1,, Thais Morata2, Simone Roggia3, Fernanda Zucki4, Benoît Pouyatos5, Thomas Venet6, Edward Krieg7, Maria Renata José8  and Adriana Lacerda9*

General comments

We thank the reviewers for their time and suggestions offered to strengthen our submission. We consider we have been able to address most comments, by restructuring the requested figures, correcting pointed errors, and modifying the Introduction and some paragraphs. Specific responses are detailed below.

Comments from reviewers

Responses from authors

Reviewer #1

The Introduction did not include a review of other or similar studies that have been reported in the literature, even though the author cited enough of such work in their discussion. Hence, the novelty of this present work is not well justified.

The introduction cites 26 papers, and covers, albeit briefly, the main aspects pertinent to the study: the epidemiology of solvent exposure and hearing risks, and clinical findings. We added a short section summarizing the literature as 1.1. Lines 56 and 111.

1.The abstract (results part) should clearly state the findings on how noise and solvents affected the auditory of the workers studied.

  1. The sentence “Workers co-exposed to solvents and noise showed significantly worse results in auditory tests in comparison with the participants in the control group” was included, but due to the maximum number of words allowed for the abstract we were unable to include more information. Lines 21 and 23.

The methodology was described in detail, but results were included in section 2.5, which should be removed and included in the results. 

Included in item 3, under table 2. Lines 245 and 252.

2. At line 91, the caption of Figure 1 should be corrected to .... "A and B", not "A e b".

2. Done. Line 127.

3. Improve the quality of Figure 2; increase the font size and restructure it to clearly show the two phases (Phase 1 and Phase 2).

3. Done. Line 163

4. Improve the quality of Figure 3; use readable/larger font and improve the quality of the lines too. 

4. Done. Line 278

5. Include a section 3.7 on solvent concentration results.

5. Done. Included in item 3, under table 2. Lines 245 and 252.

6. On page 12, the paragraph at lines 382-385 should be joined together with the preceding paragraph at lines 386-401.

6. Done. Lines 420 and 438.

Reviewer 2 Report

The topic undertaken by the authors is very interesting and important from many points of view. The article is written reasonably correctly, although it contains some errors of different severity. Therefore, in order to improve the scientific quality of the article, some corrections should be made, as listed below:

1. I don't understand what was supposed to mean and why was the first keyword "chemical" proposed?

2. Research questions, which are included in lines 41-43, should be placed at the end of the "Introduction" section, because they should result from the willingness to fill a gap in the research literature in this area.

3. The correct structure of an article should include both the "Introduction" section, which generally introduces the topic considered in the article, and the "Literature Review" section, which is a thorough review of the content contained in the current literature in a given area. There is only an "Introduction" section in the reviewed article, but it does contain some literature review.

4. The phrase "see photos of the production areas" (lines 85-86) is redundant in my opinion - just "Figure 1".

5. Are the programs mentioned in lines 86-87 standardized? Are they national programs or specific only for the analyzed companies?

6. The research questionnaire is a research tool used in the research, so firstly it should be described in more detail, and secondly - its description should not be included in the "Study Sites" subsection - because it does not apply to the places where the research was conducted.

7. I do not really understand the relationship between the numbers in the description in lines 118-122 and the numbers shown in Figure 2. Please explain this in more detail.

8. Figure 2 is of poor quality and could be a bit more elaborated.

9. The sentence in 2 lines 163-166 is a bit incomprehensible - please correct it.

10. Did the reports mentioned in line 185 cover any particular period of time? This has not been clarified.

11. Figure 4 is not legible in relation to its description; otherwise, there is no unit on the y axis.

12. I have a doubt - since the research group of people had a short work experience, may the research results be significant? 

13. In the discussion of the results, which is contained in subsection 4.1. the authors refer to only 4 literature items, the most recent of which is from 2016 - I propose to expand the discussion and support it with more recent resurces.

14. Please ensure linguistic, stylistic and punctuation correctness throughout the text (eg. line 91, caption for figure 1: is "... A e B", and probably should be "... A and B" or the sentence in lines 110-111: "Regarding noise the Brazilian limit is 85dB (A) timeweighted average (TWA), using a 5-dB exchange rate, [31];" - redundant commas, at the end of the sentence there should be a period and not a semicolon; or sentence in lines 231-234, etc.) and try to avoid unnecessary repetitions (eg. "Hearing fatigue" - lines 53-53). Other mistakes: Line 240 - the title of figure 3 should be below the figure, not above.

Author Response

Response letter of manuscript ID: 1815632

 Title: Temporary and Permanent Auditory Effects Associated With Occupational Co-Exposure to Low Levels of Noise and Solvents

Authors: Vanessa Bohn1,, Thais Morata2, Simone Roggia3, Fernanda Zucki4, Benoît Pouyatos5, Thomas Venet6, Edward Krieg7, Maria Renata José8  and Adriana Lacerda9*

General comments

We thank the reviewers for their time and suggestions offered to strengthen our submission. We consider we have been able to address most comments, by restructuring the requested figures, correcting pointed errors, and modifying the Introduction and some paragraphs. Specific responses are detailed below.

Reviewer #2

1. I don't understand what was supposed to mean and why was the first keyword "chemical" proposed?

1. We changed the order of the keywords but kept chemicals because it is usually the key word used in studies with the same theme. Other 2 reviewers did not raise this issue. Line 30.

2. Research questions, which are included in lines 41-43, should be placed at the end of the "Introduction" section, because they should result from the willingness to fill a gap in the research literature in this area.

2. Done, lines 106 and108.

3. The correct structure of an article should include both the "Introduction" section, which generally introduces the topic considered in the article, and the "Literature Review" section, which is a thorough review of the content contained in the current literature in a given area. There is only an "Introduction" section in the reviewed article, but it does contain some literature review.

3.  Done, new section added, lines 56 and 111.

4. The phrase "see photos of the production areas" (lines 85-86) is redundant in my opinion - just "Figure 1".

Done, line 121.

5. Are the programs mentioned in lines 86-87 standardized? Are they national programs or specific only for the analyzed companies?

5. Clarified in the text, lines 123 and 124.

6. The research questionnaire is a research tool used in the research, so firstly it should be described in more detail, and secondly - its description should not be included in the "Study Sites" subsection - because it does not apply to the places where the research was conducted.

6. It is detailed in section 2.4, lines 175 and 178.

7. I do not really understand the relationship between the numbers in the description in lines 118-122 and the numbers shown in Figure 2. Please explain this in more detail.

7. Figure 2 has been restructured, line 163.

8. Figure 2 is of poor quality and could be a bit more elaborated.

8. Figure 2 was redone. Line 163.

9. The sentence in 2 lines 163-166 is a bit incomprehensible - please correct it.

9.Changed, lines 199 and 202

10. Did the reports mentioned in line 185 cover any particular period of time? This has not been clarified.

10. Include year in text, lines 202 and 203

11. Figure 4 is not legible in relation to its description; otherwise, there is no unit on the y axis.

11. Figure 4 was redone, line 299. The unit was originally just mentioned in the Figure’s title.

12. I have a doubt - since the research group of people had a short work experience, may the research results be significant? 

12. The total exposure time of the workers in this study was 4.8 years. According to previous solvent studies, this duration of exposure is sufficient for the effects of such exposure to be detected. We mentioned that information in lines 527 and 531:

“Investigations which examined the effects of solvents over time indicated that hearing loss is observable two to three years earlier than is usually seen with noise exposure [2, 3, 16, 17, 18, 19]. Five or more years of noise exposure seem necessary for an effect following occupational exposures. This issue of latency is certainly dependent on the ototoxicant and the characteristics of the exposure, and needs further investigation [18, 19].

Regarding hearing fatigue, the recommended exposure of workers is 6 hours of work - https://doi.org/10.4103/1463-1741.102964.

13. In the discussion of the results, which is contained in subsection 4.1. the authors refer to only 4 literature items, the most recent of which is from 2016 - I propose to expand the discussion and support it with more recent resources.

13. Not done. The reason we were brief in section 4.1. discussing exposure duration, age and gender was that neither of these variables was associated with the studied outcomes. There are other 9 sub-sections in the discussions which discuss the findings that were associated with the outcomes.

14. Please ensure linguistic, stylistic and punctuation correctness throughout the text (eg. line 91, caption for figure 1: is "... A e B", and probably should be "... A and B" or the sentence in lines 110-111: "Regarding noise the Brazilian limit is 85dB (A) timeweighted average (TWA), using a 5-dB exchange rate, [31];" - redundant commas, at the end of the sentence there should be a period and not a semicolon; or sentence in lines 231-234, etc.) and try to avoid unnecessary repetitions (eg. "Hearing fatigue" - lines 53-53). Other mistakes: Line 240 - the title of figure 3 should be below the figure, not above

14. Done.

Reviewer 3 Report

Dear authors,

Thanks for the opportunity to review this paper. The theme is essential, usually considered by employers, and not explored in research. The study is well grounded, and the methodology is adequately described. The results follow the aim of the study and are coherent with the discussion. Limitations are adequately described. Although, my major concern is the sample size, considering the cross-sectional study design. The authors should explain the study's relevance for practice and research and the robustness of the results according to the sample size. The novelty of the study and the main findings must be highlighted. The psychometric properties of the measures should be reported.

Author Response

Response letter of manuscript ID: 1815632

Title: Temporary and Permanent Auditory Effects Associated With Occupational Co-Exposure to Low Levels of Noise and Solvents

Authors: Vanessa Bohn1,, Thais Morata2, Simone Roggia3, Fernanda Zucki4, Benoît Pouyatos5, Thomas Venet6, Edward Krieg7, Maria Renata José8  and Adriana Lacerda9*

General comments

We thank the reviewers for their time and suggestions offered to strengthen our submission. We consider we have been able to address most comments, by restructuring the requested figures, correcting pointed errors, and modifying the Introduction and some paragraphs. Specific responses are detailed below.

Reviewer #3

Although, my major concern is the sample size, considering the cross-sectional study design. The authors should explain the study's relevance for practice and research and the robustness of the results according to the sample size. The novelty of the study and the main findings must be highlighted.

Not done. The requested information can be found under 4.8, lines 519 and 554, Implications for Future Studies and Preventive Practices, and 4.9., lines 555 and 595, Study limitations and Conclusions. Not clear what additional specific information is requested.

The psychometric properties of the measures should be reported.

Not done.  We were not clear which information is needed to complement what is currently described in Table 1, line 192. The only hearing test that is not routine in the audiology clinic was Echoscan, so throught out the paper we gave more details about it than the other tests.

Round 2

Reviewer 1 Report

All the comments have been addressed, but the following minor corrections need to be addressed before it can be accepted for publications:

1. Page 2; remove the sub-heading/section "Literature Review", and section numbers "2." and "1.1". Then merge that section to the Introduction, starting as a new paragraph.

2. Page 8, line 247; correct the typo 'ehylbenzene' to 'ethylbenzene'.

3. Page 8, line 248; leave a space between the value and the units. e.g. 3.6 to 61 ng, not 3.6ng to 61ng. Make this correction throughout the manuscript. 

4. Page 8, line 250; change 'Ameri-can' to 'American'. Provide reference citations for the Brazilian exposure limits and ACGIH at line 251.

Author Response

Response letter of manuscript ID: 1815632

August 5th 2022

Title: Temporary and Permanent Auditory Effects Associated With Occupational Co-Exposure to Low Levels of Noise and Solvents

Reviewer 1

  1. Page 2; remove the sub-heading/section "Literature Review", and section numbers "2." and "1.1". Then merge that section to the Introduction, starting as a new paragraph.
  • Done, pg 2. line 55
  1. Page 8, line 247; correct the typo 'ehylbenzene' to 'ethylbenzene'.
  • Done, pg 8, line 247
  1. Page 8, line 248; leave a space between the value and the units. e.g. 3.6 to 61 ng, not 3.6ng to 61ng. Make this correction throughout the manuscript. 
  • Done Page 8, line 248, as well as throughout the document.
  1. Page 8, line 250; change 'Ameri-can' to 'American'. Provide reference citations for the Brazilian exposure limits and ACGIH at line 251.
  • Done Page 8, line 250 and 251;
  1. English language and style are fine/minor spell check required
  • Done

Reviewer 3 Report

Thanks for the opportunity to review this new version of the paper.

The authors have made a great effort in this revision, turning the information more transparent, clear, and coherent among the several sections of the manuscript. I believe that with the changes introduced in the manuscript, it becomes more robust, theoretically and methodologically, highlighting the originality of the study, the main findings, and implications for clinical practice and occupational health. 

Author Response

Response letter of manuscript ID: 1815632

August 5th 202

Title: Temporary and Permanent Auditory Effects Associated With Occupational Co-Exposure to Low Levels of Noise and Solvents

Reviewer 3

The authors have made a great effort in this revision, turning the information more transparent, clear, and coherent among the several sections of the manuscript. I believe that with the changes introduced in the manuscript, it becomes more robust, theoretically and methodologically, highlighting the originality of the study, the main findings, and implications for clinical practice and occupational health.

  1. English language and style are fine/minor spell check required
  • Done